# Transgenerational Effects of Di(2-Ethylhexyl) Phthalate on Anogenital Distance, Sperm Functions and DNA Methylation in Rat Offspring

**DOI:** 10.3390/ijms22084131

**Published:** 2021-04-16

**Authors:** Ping-Chi Hsu, Jia-Ying Jhong, Li-Ping Huang, Kuo-Hsin Lee, Hsin-Pao Chen, Yue-Leon Guo

**Affiliations:** 1Department of Safety, Health and Environmental Engineering, National Kaohsiung University of Science and Technology, Kaohsiung 81157, Taiwan; pchsu@nkust.edu.tw (P.-C.H.); pchsu0501@gmail.com (J.-Y.J.); 2Department of Nursing, Chung-Jen College of Nursing, Health Sciences and Management, Chiayi 62241, Taiwan; m108@cjc.edu.tw; 3Department of Emergency Medicine, E-Da Hospital, School of Medicine, College of Medicine, I-Shou University, Kaohsiung 82445, Taiwan; peter1055@gmail.com; 4Division of Colon and Rectal Surgery, Department of Surgery, E-Da Hospital, School of Medicine, College of Medicine, I-Shou University, Kaohsiung 82445, Taiwan; ed102430@edah.org.tw; 5Environmental and Occupational Medicine, National Taiwan University (NTU) College of Medicine and NTU Hospital, Taipei 100229, Taiwan; 6Institute of Environmental and Occupational Health Sciences, National Taiwan University College of Public Health, Taipei 100025, Taiwan

**Keywords:** anogenital distance, di(2-ethylhexyl) phthalate, epigenetic, sperm DNA methylation, sperm function, transgenerational effect

## Abstract

Di(2-ethylhexyl) phthalate (DEHP) is widely used as a plasticizer in the manufacture of polyvinylchloride plastics and has been associated with concerns regarding male reproductive toxicity. In this study, we hypothesized that maternal exposure to DEHP induces transgenerational inheritance of adult-onset adverse reproductive outcomes through the male germline in the F1, F2, and F3 generations of male offspring. Pregnant rats were treated with 5 or 500 mg of DEHP/kg/day through gavage from gestation day 0 to birth. The offspring body weight, anogenital distance (AGD), anogenital index (AGI), sperm count, motility, and DNA fragmentation index (DFI) were measured for all generations. Methyl-CpG binding domain sequencing was performed to analyze sperm DNA methylation status in the F3. DEHP exposure at 500 mg/kg affected AGD, AGI, sperm count, mean DFI, and %DFI in the F1; AGD, sperm count, and mean DFI in the F2; and AGD, AGI, mean DFI, and %DFI in the F3. DEHP exposure at 5 mg/kg affected AGD, AGI, sperm count, and %DFI in the F1; sperm count in the F2; and AGD and AGI in F3. Compared with the control group, 15 and 45 differentially hypermethylated genes were identified in the groups administered 5 mg/kg and 500 mg/kg DEHP, respectively. Moreover, 130 and 6 differentially hypomethylated genes were observed in the groups administered 5 mg/kg and 500 mg/kg DEHP. Overall, these results demonstrated that prenatal exposure to DEHP caused transgenerational epigenetic effects, which may explain the observed phenotypic changes in the male reproductive system.

## 1. Introduction

Di(2-ethylhexyl) phthalate (DEHP), a well-known endocrine-disrupting chemical (EDC), is widely used as a plasticizer in the manufacture of polyvinylchloride (PVC) plastics and has been associated with concerns about antiandrogenic activity and male reproductive toxicity [1,2,3]. DEHP and its metabolites are often detected in human urine and amniotic fluid, which underscores its risk as a toxic hazard [4,5] and indicates potential effects on fetal development [6,7,8].

Anogenital distance (AGD), an androgen-sensitive marker of development in humans and rodent models that reflects gestational androgen exposure has been used to study the effects of prenatal exposure to various potential EDCs [9,10]. Furthermore, the anogenital index (AGI) was developed to adjust for the effect of body weight on AGD in animal studies [11]. Stenz et al. [12] revealed a significant reduction in AGD in male mouse offspring exposed to 300 mg/kg/day DEHP during embryonic days 9 to 19. In epidemiological studies, the relationships of DEHP exposure with AGD and AGI remain controversial [13]. Maternal urinary concentrations of DEHP metabolites were associated with reduced AGD in 85 boys in the United States [5]. Higher urinary concentration of mono-2-ethylhexyl phthalate in 111 pregnant Japanese women was significantly associated with reduced AGI in their male offspring [7]. By contrast, no significant association was observed between the phthalate level in the amniotic fluid or urine of Taiwanese women and AGD in their 33 male newborns [14]. These findings indicate that DEHP exposure may disrupt the development of human male genitals; however, studies examining the concentrations of phthalate metabolites in urine or amniotic fluid and AGD in male offspring have yielded inconsistent results.

Epigenetic mechanisms including DNA methylation, histone modification, and expression of noncoding regulatory RNA are responsible for inherited changes in gene expression during embryogenesis and early development [15]. Among the epigenetic mechanisms involving exposure to environmental chemicals and epigenetic regulation, DNA methylation has been investigated the most extensively and is the most accurately characterized [16,17]. Environmental EDCs have been linked to aberrant alterations of epigenetic pathways in experimental and epidemiological studies as reviewed in [18,19,20]. Evidence supporting the effect of DEHP exposure during the fetal window of susceptibility on sperm DNA methylation is accumulating. Prados et al. [21] demonstrated that prenatal DEHP exposure may promote alterations of sperm DNA methylation and may be associated with reduced sperm count in C57BL/6J mice. Prenatal DEHP exposure induced long-lasting and robust promoter methylation-related silencing of fundamental genes in spermatozoa [12]. Furthermore, in utero DEHP exposure was associated with enhanced DNA methylation for genes involved in androgen response, estrogen response, and spermatogenesis [22].

In two animal models, in utero exposure to a plastic compound mixture (i.e., DEHP, bisphenol A, and dibutyl phthalate) demonstrated the potential to promote transgenerational epigenetic inheritance of reproductive abnormalities [23,24]. Another rat model confirmed that 700 mg/kg DEHP exposure during the critical time for embryonic development promoted the expression of DNA methyltransferase enzyme, which in turn caused changes in the genomic imprinting methylation pattern and may have induced epigenetic transgenerational inheritance of cryptorchidism in male offspring [25]. However, limited data are available from animal studies to indicate that maternal exposure to low or high DEHP doses promotes transgenerational inheritance of adult-onset adverse reproductive outcomes through the male germline in male offspring. The present study had the following aims: first, we intended to establish an experimental animal model for investigating the transgenerational effects of low and high DEHP dose exposures on development and sperm function in the F1, F2, and F3 generations. Second, we aimed to evaluate the alterations of sperm DNA methylation in the F3 generation to obtain evidence of dose–response characteristics and a potential epigenetic mechanism of action relevant to male reproductive health.

## 2. Results

### 2.1. Effects of DEHP Exposure on Body Weight, AGD, and AGI

In the F1 generation, prenatal exposure to the vehicle, 5 mg/kg/day DEHP, or 500 mg/kg/day DEHP from gestational day (GD) 0 to GD 18 did not result in significant differences in the body weights of offspring rats (Figure 1A). The AGD of the 500 mg/kg DEHP group was significantly lower than those of the 5 mg/kg DEHP and control groups at postnatal day (PND) 22, 28, 31, 34, 37, 67, and 70 (*p* < 0.05; Figure 1B). The AGD of the 5 mg/kg DEHP group was significantly lower than those of the control groups at PND 25, 28, 31, and 34 (*p* < 0.05; Figure 1B). The AGI of the 500 mg/kg DEHP group was significantly lower than those of the 5 mg/kg DEHP and control groups at PND 22, 25, 28, 31, 34, 37, 40, 61, 64, and 67 in the F1 generation (*p* < 0.05; Figure 1C). However, in the F1 generation, no significant differences were observed in the AGI between the 5 mg/kg DEHP and control groups.

In the F2 generation, the body weights of rats in the three groups did not differ significantly (Figure 2A). The AGD of the 500 mg/kg DEHP group was significantly lower than that of the control group at PND 22, 58, 61, 64, 67, and 70 (*p* < 0.05) (Figure 2B). However, the AGD of the 5 mg/kg DEHP group was significantly higher than those of the control and 500 mg/kg DEHP groups at PND 28, 31, 34, 40, 43, 46, 49, 52, 55, 61, 64, 67, and 70 (*p* < 0.05; Figure 2B). Similarly, the AGI of the 5 mg/kg DEHP group was significantly higher than those of the control and 500 mg/kg DEHP groups at PND 28, 43, 46, 49, 61, 64, 67, and 70 (*p* < 0.05; Figure 2C).

In the F3 generation, the body weights of rats in the three groups did not differ significantly (Figure 3A). The AGD of the 500 mg/kg DEHP group was significantly lower than those of the 5 mg/kg DEHP and control groups for PND 28 to 70 (*p* < 0.05; Figure 3B). The AGI of the 500 mg/kg DEHP group was significantly lower than those of the 5 mg/kg DEHP and control groups from PND 34 to 70 (*p* < 0.05; Figure 3C). Furthermore, the AGD of the 5 mg/kg DEHP group was significantly lower than that of the control group on PND 64, 67, and 70 (*p* < 0.05; Figure 3B). The AGI of the 5 mg/kg DEHP group was significantly lower than that of the control group on PND 49 (*p* < 0.05; Figure 3C).

### 2.2. Effects of DEHP Exposure on Sperm Count and Motility

In the F1 and F2 generations, the 5 mg/kg and 500 mg/kg DEHP groups had lower sperm counts than the control group (*p* < 0.05; Figure 4A). However, in the F3 generation, no effect on sperm count was observed (Figure 4A). In the F1 to F3 generations, DEHP exposure did not affect sperm motility (Figure 4B).

### 2.3. Effect of DEHP Exposure on Sperm Chromatin DNA

Epididymis spermatozoa were analyzed using a flow cytometry (FCM) sperm chromatin structure assay (SCSA) to investigate DEHP-induced damage to DNA chromatin integrity. In the F1 to F3 generations, the mean DNA fragmentation index (DFI) was significantly higher in the 500 mg/kg DEHP group than in the 5 mg/kg DEHP and control groups. However, no significant difference in mean DFI was observed between the 5 mg/kg DEHP and control groups (Figure 5A).

Figure 5B illustrates the effects of DEHP exposure on the percentage of spermatozoa with DNA damage (%DFI). In the F1 generation, %DFI was significantly higher in the 5 mg/kg DEHP and 500 mg/kg DEHP groups than in the control group (*p* < 0.05; Figure 5B). In the F2 generation, DEHP exposure did not affect %DFI (Figure 5B). In the F3 generation, %DFI was significantly higher in the 500 mg/kg DEHP group than in the 5 mg/kg DEHP and control groups. However, no significant difference in %DFI was observed between the 5 mg/kg DEHP and control groups (Figure 5B).

### 2.4. Hypermethylated and Hypomethylated Genes in Sperm DNA

After normalization (capture/input), hypermethylated genes were screened in the 500 mg/kg and 5 mg/kg DEHP groups. The DEHP treatments were compared with the control group, with hypermethylation defined as a change of at least twofold in the reads per kilobase per million (RPKM) in the F3 generation. In the F3 generation, 15 and 45 differentially hypermethylated genes were identified in the 5 mg/kg DEHP group (Figure 6A) and 500 mg/kg DEHP group (Figure 6B), respectively, relative to the control group; these genes were related to upregulation of biological processes (BPs) in the gene ontology (GO) classification. In the 5 mg/kg DEHP group, hypermethylated genes were focused on spermatogenesis and intracellular signal transduction (Figure 6B).

After normalization (capture/input), hypomethylated genes were screened in the 500 mg/kg and 5 mg/kg DEHP groups. The DEHP treatments were compared with the control group, with hypomethylation defined as a change of less than half-fold in RPKM in the F3 generation. Furthermore, 130 and 6 differentially hypomethylated genes were identified in the 5 mg/kg DEHP group (Figure 7A) and 500 mg/kg DEHP group (Figure 7B), respectively, relative to the control group; these genes were related to the downregulation of BPs in the GO classification. In the 5 mg/kg DEHP group, hypomethylated genes were focused on spermatogenesis and the negative regulation of apoptotic processes (Figure 7A). In the 500 mg/kg group, hypomethylated genes were focused on the negative regulation of T-cell activation, vesicles for transport from the endoplasmic reticulum to the Golgi apparatus, and positive regulation of the JNK cascade (Figure 7B).

### 2.5. Gene Symbol, Gene Name, Chromosome, and Ratio of Relative Fold Expression of Hypermethylated Genes

Table 1 summarizes the 10 genes with the highest hypermethylation for the 500 mg/kg DEHP, 5 mg/kg DEHP, and control groups in the F3 generation. The fold changes of RPKM were scored for the 5 mg/kg DEHP and control groups, and this score was multiplied by the relative fold changes of RPKM in the 500 mg/kg DEHP and 5 mg/kg DEHP groups. The results revealed that, in a dose–response manner, the 10 most significant hypermethylated genes were *FAM222A*, *GAPDH*, *CPSF2*, *ESRRA*, *ANKRD13D*, *TTBK1*, *TBCCD1*, *FAM220A*, *CD302*, and *MTMR7* (Table 1).

### 2.6. Gene Symbol, Gene Name, Chromosome, and Ratio of Relative Fold Expression of Hypomethylated Genes

Table 2 summarizes the 10 genes with the greatest hypomethylation for the 500 mg/kg DEHP, 5 mg/kg DEHP, and control groups in the F3 generation. The fold changes in RPKM were scored for the 5 mg/kg DEHP and control groups, and this score was multiplied by the relative fold changes of RPKM in the 500 mg/kg and 5 mg/kg DEHP groups. The results revealed that, in a dose–response manner, the 10 most significant hypomethylated genes were *ATG1611*, *FAM13A*, *ATOH7*, *BCAP29*, *MRPS18B*, *DUSP22*, *AOX2*, *RBM*, *MED24*, and *DDX19A* (Table 2).

## 3. Discussion

This systematic measurement of the effects of DEHP in the F1, F2, and F3 generations of male rats revealed a novel toxicological mechanism of epigenetic transgenerational inheritance. The results indicate that exposure to DEHP during pregnancy disrupts sperm function across generations through paternal lineage. In the F1 generation, 500 mg/kg DEHP exposure affected AGD, AGI, sperm count, mean DFI, and %DFI; in the F2 generation, it affected AGD, sperm count, and mean DFI; and in the F3 generation, it affected AGD, AGI, mean DFI, and %DFI. In the F1 generation, 5 mg/kg DEHP exposure affected AGD, AGI, sperm count, and %DFI; in the F2 generation, it affected sperm count; and in the F3 generation, it affected AGD and AGI. Specifically, for the 5 mg/kg DEHP group, AGD and AGI were significantly higher in the F2 generation than in the F1 generation. Furthermore, this study was the first to compare the detailed epigenetic transgenerational effects of prenatal exposure to low and high doses of DEHP on the male reproductive system.

We found that 500 mg/kg DEHP exposure significantly lowered AGD and AGI in the F1, F2, and F3 generations compared with the control. Several studies have indicated that DEHP can cause a significant reduction in male AGD. Significantly decreased AGD was observed in rats with exposure to 750 mg/kg/day DEHP from GD 13 to 20, whereas exposure to 750 mg/kg/day DEHP from PND 23 to 53 induced slight but significant reductions in male AGD [9]. Prenatal exposure of C57BL/6J mice to 300 mg/kg/day DEHP from GD 9 to 19 reduced AGD, sperm count, and sperm motility [12]. AGD is a stable anatomical biomarker that reflects androgen action during the fetal testis development period in rodents as well as in humans [10]. Specifically, the AGD and AGI of the 5 mg/kg DEHP group were significantly higher than those in the control and 500 mg/kg DEHP groups in the F2 generation. EDCs may have nonmonotonic or U-shaped dose–response curves [26]. Thus, a low or specific concentration may have the reverse action of a higher dose.

Notably, the effects of low-dose and high-dose DEHP exposure differed between generations. This was likely because during prenatal DEHP exposure, the F1, F2, and F3 generations received different levels of DEHP during specific developmental periods. The F1 generation was exposed to DEHP as a developing pup; the F2 generation was exposed to DEHP during testis development; and the F3 generation was the first generation without direct exposure to DEHP.

Several studies have indicated that prenatal exposure to DEHP might induce developmental toxicity, endocrine disruption, and reproductive hazards in a dose–response manner in male offspring [27,28]. However, few studies have reported the transgenerational effects of low-dose and high-dose DEHP exposure on male reproduction. The present study is the first to demonstrate that at 5 mg/kg and 500 mg/kg, DEHP exposure significantly increased the mean DFI in the F1, F2, and F3 generations. Sperm count was significantly reduced in the F1 and F2 generations. Notably, in the F3 generation, the hypermethylated and hypomethylated genes in the 500 mg/kg DEHP group were focused on spermatogenesis. Doyle et al. [29] revealed that in utero DEHP exposure (500 mg/kg) delayed the onset of puberty and decreased sperm count in the F3 generation of male CD-1 mice, as well as increasing the number of abnormal seminiferous tubules in the F3 and F4 generations. In another animal study, pregnant Sprague–Dawley (SD) rats were administered dibutyl phthalate (500 mg/kg) through oral gavage from GD 8 to 14; this reduced sperm count and Sertoli cell count in the F1, F2, and F3 generations [30]. These transgenerational effects were observed at doses higher than those investigated for human exposure. Thus, the environmentally relevant dose should be considered when determining the transgenerational impacts of DEHP on male reproduction in future studies.

Few studies have investigated the transgenerational effects of DEHP on hypermethylated and hypomethylated genes. The present study focused on the 10 most significant hypermethylated and hypomethylated genes that have a dose–response relationship with DEHP. Family with sequence similarity 222, member A (*FAM222A*) was the most hypermethylated gene, with the highest score for low- and high-dose DEHP exposure. Recent studies have reported that *FAM222A* as a putative brain atrophy susceptibility gene and that the protein encoded by *FAM222A* is pathologically relevant in Alzheimer disease and Nasu–Hakola disease [31,32]. Moreover, glyceraldehyde-3-phosphate dehydrogenase (*GAPDH*) was the second most hypermethylated gene; the expression and compartmentalization of the glycolytic enzymes of *GAPDH* as well as pyruvate kinase are involved in boar spermatogenesis [33]. Furthermore, cleavage and polyadenylation specific factor 2 (*CPSF2*), the third most hypermethylated gene, encodes a sperm protein that can modulate gene expression in human spermatozoa [34].

Autophagy Atg16L1 (*ATG16L1*) was the most hypomethylated gene according to its equivalent score. The deletion of *ATG16L1* enhances *NLRP3*, possibly through mitochondrial impairment [35]. The second most hypomethylated gene was family with sequence similarity 13, member A (*FAM13A*), which has been identified as a marker gene in insulin sensitivity and lipolysis. Human *FAM13A* is highly expressed in adipose tissue, the duodenum, the placenta, and the thyroid [36].

Our previous studies have revealed that the concentration of DEHP in the personal breathing zone of an employee’s workstation is positively associated with sperm DNA damage and negatively correlated with sperm motility [37] and that the urinary metabolites of DEHP are associated with sperm motility, reactive oxygen species (ROS) generation, and apoptosis in those who work with PVC [38]. Moreover, our previous animal studies have revealed that low-dose DEHP exposure during adolescence increases the apoptosis rate of spermatocytes, testis atrophy, and the sperm DFI, as well as elevating ROS levels and exacerbating 1,2-dimethylhydrazine-induced colon tumorigenesis [3,39]. Although DEHP-related carcinogenicity was not investigated in the present experiments, the present finding that epigenetic modifications of hypermethylated and hypomethylated genes are related to DEHP exposure in a dose–response manner suggests that prenatal DEHP exposure may epigenetically affect male reproduction through transgenerational effects.

## 4. Materials and Methods

### 4.1. Study Design

Figure 8 summarizes the experimental procedures of this transgenerational experimental design, including the breeding strategy and timeline of individual measurements. Pregnant SD rats (F0) were treated through gavage on GD 0 to birth with a vehicle control (corn oil), 5 mg/kg/day DEHP, or 500 mg/kg/day DEHP. On PND 21, three male offspring were randomly selected from each litter; in total, nine offspring (F1) were obtained from each group. Male rats born to mice from the F1 generation were labeled the F2 generation. Male rats born to mice from the F2 generation were labeled the F3 generation.

Body weight, AGD, and AGI were measured every 3 days from PND 22 to 70. On PND 84, the rats were anesthetized through CO_2_ inhalation and the left epididymal sperm suspension was analyzed for sperm functions, including sperm count, sperm motility, and sperm chromatin SCSA in the F1, F2, and F3 generations. Methyl-CpG binding domain sequencing (MBD-seq) was performed to analyze DNA methylation status in the F3 generation (Figure 8).

### 4.2. Animals and Treatment

All experimental procedures for animal care, treatment, euthanasia, and tissue collections were approved by the Institutional Animal Care and Use Committee of the National Kaohsiung University of Science and Technology (NKUST) (approval number 105–002). The NKUST Environmental and Occupational Safety and Health Center approved the protocols for the use of environmental chemicals. SD rats were obtained from the BioLASCO Experimental Animal Center (Taipei, Taiwan). Rats were maintained in an animal chamber with a temperature 24–26 °C and a 12 h light/dark photoperiod. The humidity in the chamber was maintained at 55%–60%. The rats were given a standard diet and distilled water ad libitum. The first appearance of a vaginal plug was used to define GD 0. Pregnant rats (three per group) were treated with 5 mg/kg/day DEHP, 500 mg/kg/day DEHP, or the corn oil vehicle control through gavage from GD 0 until they gave birth; these dosages were selected according to the lowest-observed-adverse-effect level for effects on male reproductive development [40] and sperm dysfunction [41] in offspring. DEHP (99.5% purity, CAS no. 117-81-7) and corn oil were purchased from Sigma-Aldrich (St. Louis, MO, USA).

### 4.3. Breeding the F1, F2, and F3 Generations

The pregnant rats treated with DEHP or corn oil were designated the F0 generation. The offspring of the F0 generation were the F1 generation (Figure 8). To study the transmission of DEHP effects over multiple generations through the male germline, three male F1 offspring were randomly selected from the litters of each of the treatment and control groups and naturally mated with nonexposed female rats of proven fertility to obtain the F2 generation. Three male F2 rats were selected from each the litters of each of the treatment and control groups for breeding to obtain the F3 generation.

### 4.4. Body Weight, AGD, and AGI

Body weight, AGD, and AGI were recorded every 3 days from PND 22 to PND 72 in the F1, F2, and F3 generations. For all male offspring, AGD was determined by measuring the distance from the center of the anus to posterior edge of the genital papilla by using a digital caliper and a dissecting microscope equipped with an eyepiece reticle (Zeiss; Stemi, 1000-C, Göttingen, Germany). This examination was performed by a single investigator who was unaware of the animals’ exposure levels.

### 4.5. Sperm Count and Motility Analysis

The right cauda epididymis of each control and treated rat were removed and quickly transferred to a fresh tube with 1 mL of human tubal fluid medium. This medium was maintained at 34 °C in an environment saturated with 5% CO_2_. After 5 min, the cauda epididymis was minced using curved scissors and the sperm was dispersed into the medium. A 1:10 dilution of the sperm suspension was prepared; then, 10 μL of sperm suspension was placed on a hemocytometer chamber and examined under a phase contrast microscope (Olympus CH2, Tokyo, Japan) for determining the sperm count and motility was expressed as the ratio between the number of motile sperm and the total number of sperm.

### 4.6. Sperm Chromatin Structure Assay

The SCSA was performed to assess the integrity of sperm DNA. Sperm aliquots (0.1 mL) diluted to a concentration of 1 × 10^6^ cells/mL were mixed with 0.2 mL of Lysis solution (0.1% Triton X-100, 0.08 N HCl, and 0.15 M NaCl; pH 1.2). Acridine orange (AO) is a staining solution used to distinguish double-stranded nucleic acids from stained single-stranded ones. The spermatozoa were stained by adding 0.3 mL of AO solution (2 mg/mL AO, 0.15 M NaCl, 126 mM Na_2_HPO_4_, 1 mM ethylenediaminetetraacetic acid, and 37 mM citric acid buffer; pH 7.4). AO is used to distinguish stained double-stranded nucleic acids from those that are single-stranded. In SCSA analysis using BD FACS-Canto II FCM (BD Biosciences, San Jose, CA, USA), each AO-stained spermatozoon DNA analyzed produced green fluorescence (detected by a 515–530 nm band-pass filter) and red fluorescence (detected by a 630–650 nm long-pass filter). Sperm DNA DFI and %DFI were analyzed according to Evenson and Wixon (2005) [42]; these variables relate to the susceptibility of sperm to chromatin DNA damage.

### 4.7. Sperm DNA Extraction

For DNA extraction in the F3 generation, the epididymis was dissected and placed on a Petri dish with a droplet of phosphate-buffered saline. The sample was then transferred to an Eppendorf tube, and the fragments were allowed to sediment for 30 min at 37 °C. The final supernatant was carefully removed and centrifuged at 6000× *g* for 3 min to obtain a sperm pellet. Sperm samples were lysed through beating with glass beads. Genomic DNA was then extracted using a WelPrep DNA kit (Welgene Biotech, cat no. D001, Taiwan) in accordance with the manufacturer’s instructions.

### 4.8. Methyl-CpG Binding Domain Sequencing

Three experimental pools were generated for the control and DEHP-treated lineages in the F3 generation; each pool contained sperm-extracted DNA from nine animals each from three litters. A total of 1 µg of total DNA was sonicated using the Covaris M220 instrument to obtain fragments ranging from 180 to 280 bps. The DNA size was determined using TapeStation D1000 ScreenTape (Agilent Technologies, CA, USA). Methylated DNA was MBD enriched using an EpiMark Methylated DNA Enrichment Kit (New England BioLabs, MA, USA). Enriched methylated DNA was end-repaired, A-tailed, and adaptor-ligated following the sample preparation protocol of TruSeq DNA (Illumina, CA, USA). The sequences of genomic DNA fragments were determined using NextSeq500 (Illumina) through single-end sequencing with a read length of 75 bp. MBD-seq was performed by Welgene Biotech Co., Ltd. (Taipei, Taiwan) in accordance with the manufacturer’s instructions (Illumina).

The raw sequences were then filtered to obtain qualified reads. The Trimmomatic tool was used to trim or remove reads depending on their quality score [43]. The qualified reads were aligned with the rat reference genome sequence (RN5.0), which was retrieved from the University of California San Francisco (UCSF) database by using the Burrows–Wheeler transform [44]. After applying Burrows–Wheeler alignment mapping, methylated DNA immunoprecipitation (MeDIP) was used to analyze the enrichment scores and determine the differences in methylation between the samples based on a window size of 500 bp. The enrichment score denotes the CpG enrichment within the given region relative to the reference genome. For short reads obtained by sequencing nonenriched DNA fragments (input experiments), the enrichment values should be close to 1. By contrast, an MBD/MeDIP-seq experiment should return sequences with high CpG enrichment scores. Reads within 5000 bp upstream of genes were used to calculate the RPKM [45]. RPKM filtering was performed for each group by using customized criteria.

### 4.9. Analysis of Hypermethylation and Hypomethylation in Sperm DNA

The genome was divided into 500-bp windows, and the methylation level of each window was quantified. The nearest upstream and downstream genes from each window were further annotated. After normalization (capture/input) for the DEHP treatment groups, RPKM values with a greater than twofold change relative to the control group were considered to represent hypermethylation, and RPKM values with a less than half-fold change were considered to represent hypomethylation.

Hypermethylated genes:

((RPKM of capture/input in 5 mg/kg DEHP or 500 mg/kg DEHP group)/(RPKM of capture/input in control group)) > 2.

Hypomethylated genes:

((RPKM of capture/input in 5 mg/kg DEHP or 500 mg/kg DEHP group)/(RPKM of capture/input in control group)) < 0.5.

The obtained hypermethylated and hypomethylated genes were added to the Database for Annotation, Visualization, and Integrated Discovery for reuse; then, the genes were grouped according to the BPs of GO.

The 10 most hypermethylated and hypomethylated genes that had a dose–response relationship with DEHP were calculated as follows:

Relative fold of RPKM expression between 5 mg/kg DEHP and control groups × Relative fold of RPKM expression between 500 mg/kg DEHP and 5 mg/kg DEHP groups.

### 4.10. Statistical Analysis

One-way analysis of variance (ANOVA) was performed for each group to identify significant differences in means between control and DEHP-exposed groups; ANOVA was executed using the JMP statistical package (version 10.0; SAS Institute Inc., Gary, NC, USA). The data of multiple male pups originating from the same litter were averaged and combined, and data from three litters were used for every treatment group in each generation. All data are expressed herein as the mean ± standard error of the mean. When ANOVA results were significant (*p <* 0.05), Tukey’s test was used for post hoc analysis.

## 5. Conclusions

This study revealed that prenatal exposure to low-dose DEHP caused transgenerational epigenetic effects in a well-established animal model, and such effects might explain phenotypic changes in the male reproductive system. Further analyses are required to identify specific alternations in sperm DNA methylation and sperm dysfunction, which would deepen the current understanding of this novel observation.

## Figures and Tables

**Figure 1 ijms-22-04131-f001:**
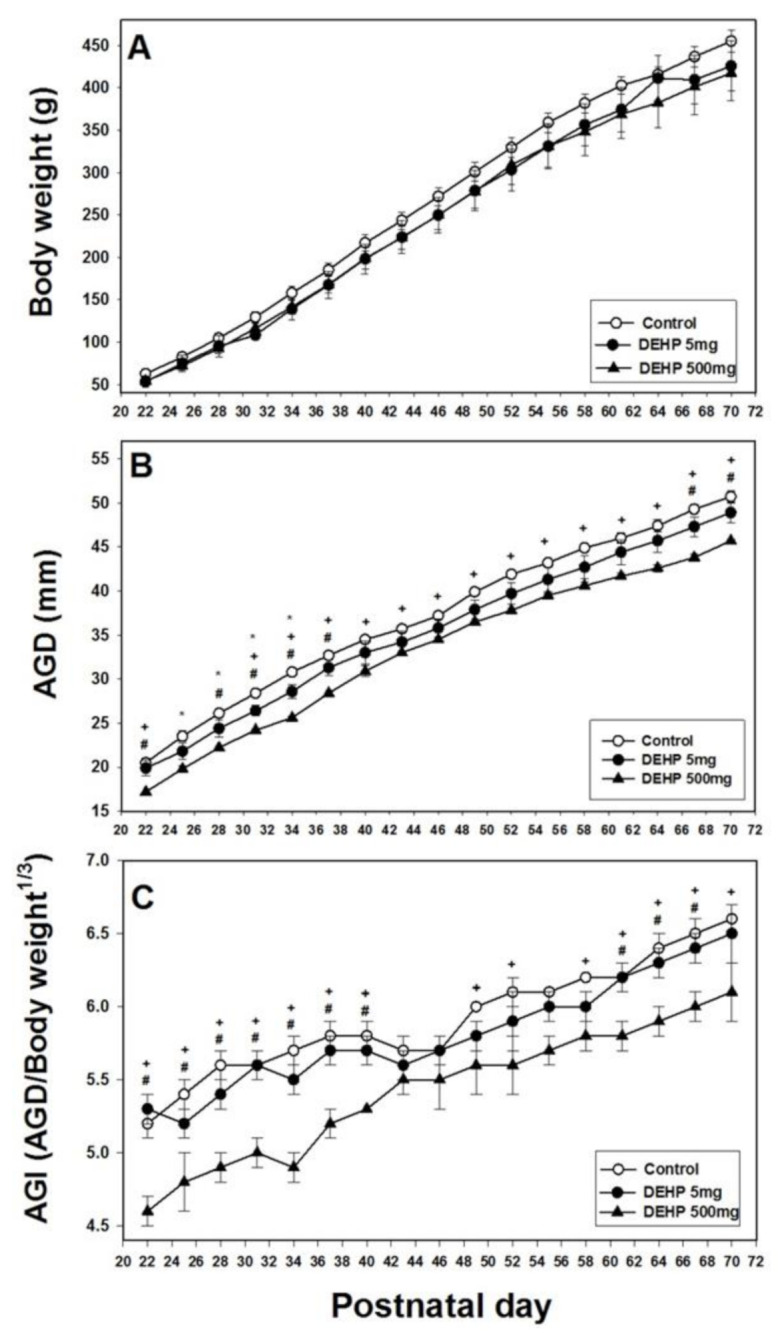
Body weight (**A**), anogenital distance (AGD) (**B**), and anogenital index (i.e., weight-adjusted AGD) (**C**) in male offspring following prenatal exposure to 5 mg/kg DEHP, 500 mg/kg DEHP, or corn oil from postnatal day 22 to 70 for the F1 generation. Data from three litters were used for every treatment group. Error bars represent the standard error. For the curves, * indicates a significant difference between the 5 mg/kg DEHP and control groups; + indicates a significant difference between the 500 mg/kg DEHP and control groups; and # indicates a significant difference between the 5 mg/kg and 500 mg/kg DEHP groups.

**Figure 2 ijms-22-04131-f002:**
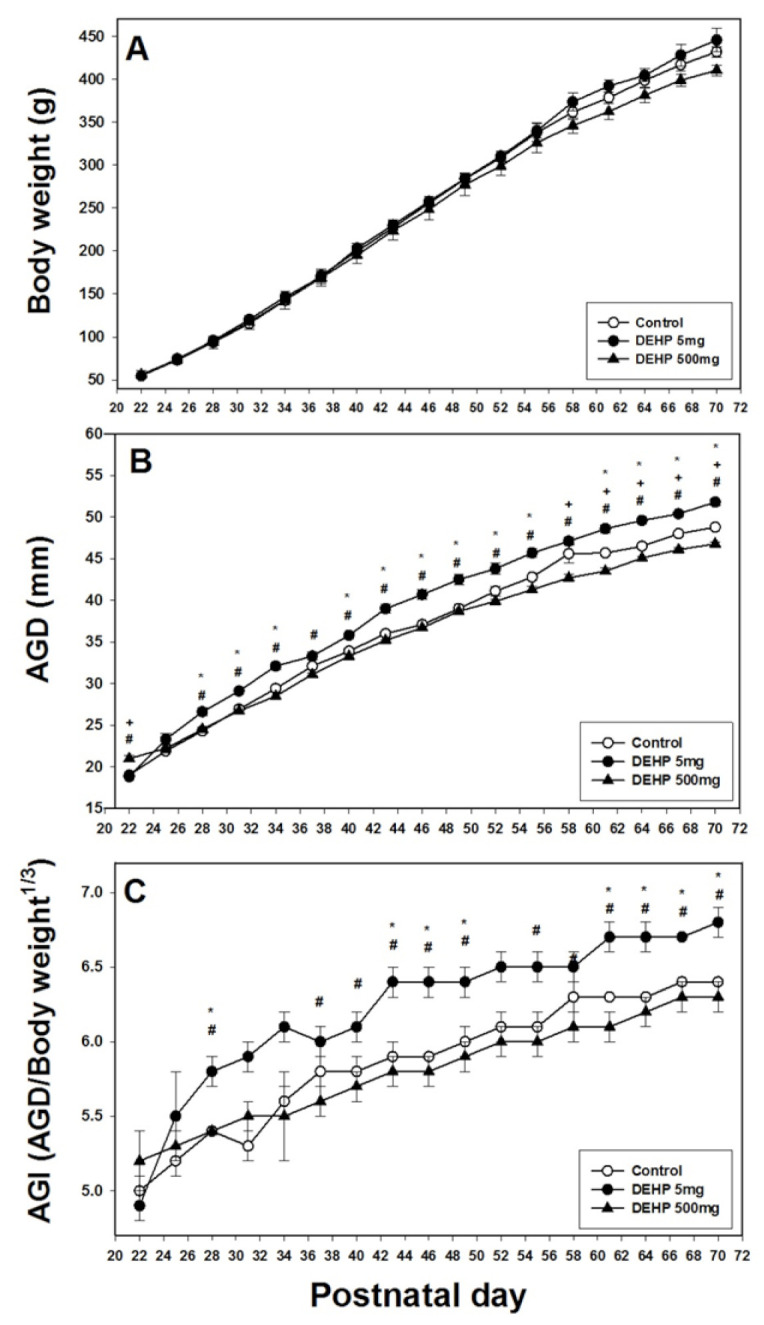
Body weight (**A**), anogenital distance (AGD) (**B**), and anogenital index (i.e., weight-adjusted AGD) (**C**) in male offspring following prenatal exposure to 5 mg/kg DEHP, 500 mg/kg DEHP, or corn oil from postnatal day 22 to 70 for the F2 generation. Data from three litters were used for every treatment group. Error bars represent the standard error. For the curves, * indicates a significant difference between the 5 mg/kg DEHP and control groups; + indicates a significant difference between the 500 mg/kg DEHP and control groups; and # indicates a significant difference between the 5 mg/kg and 500 mg/kg DEHP groups.

**Figure 3 ijms-22-04131-f003:**
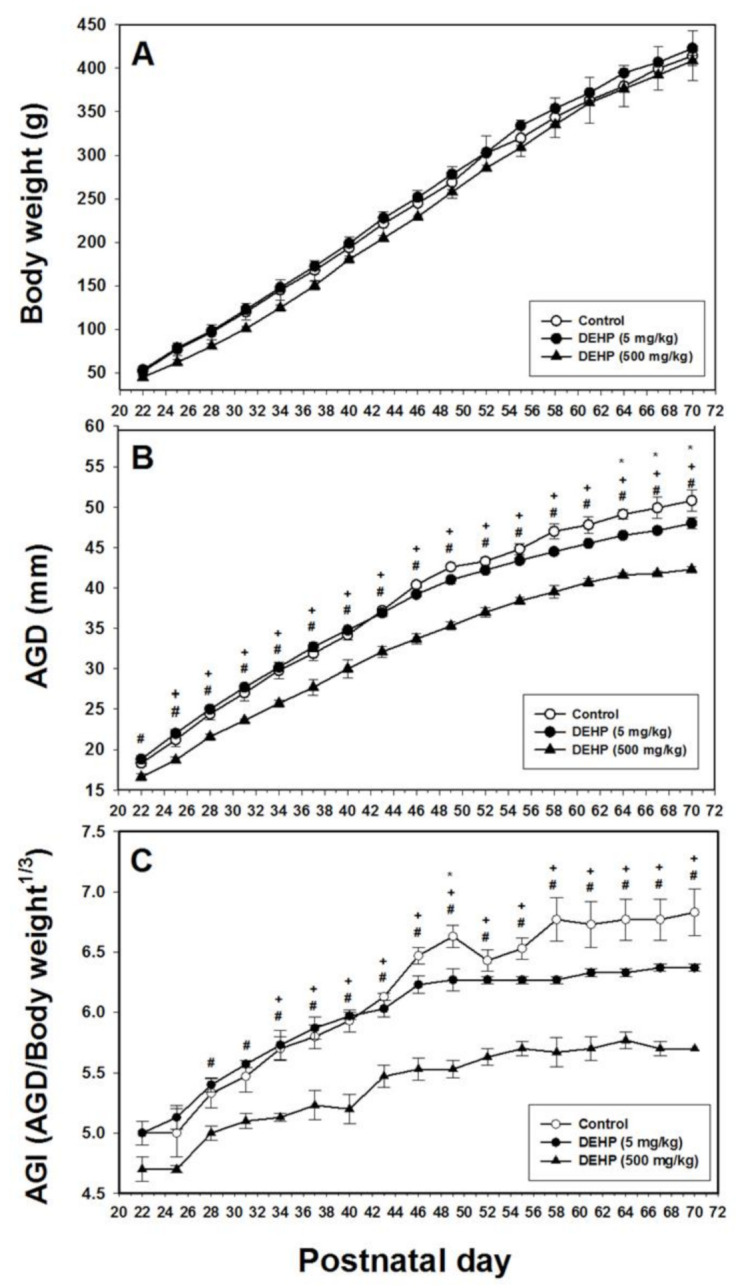
Body weight (**A**), anogenital distance (AGD) (**B**), and anogenital index (i.e., weight-adjusted AGD) (**C**) in male offspring following prenatal exposure to 5 mg/kg DEHP, 500 mg/kg DEHP, or corn oil from postnatal day 22 to 70 for the F3 generation. Data from three litters were used for every treatment group. Error bars represent the standard error. For the curves, * indicates a significant difference between the 5 mg/kg DEHP and control groups; + indicates a significant difference between the 500 mg/kg DEHP and control groups; and # indicates a significant difference between the 5 mg/kg and 500 mg/kg DEHP groups.

**Figure 4 ijms-22-04131-f004:**
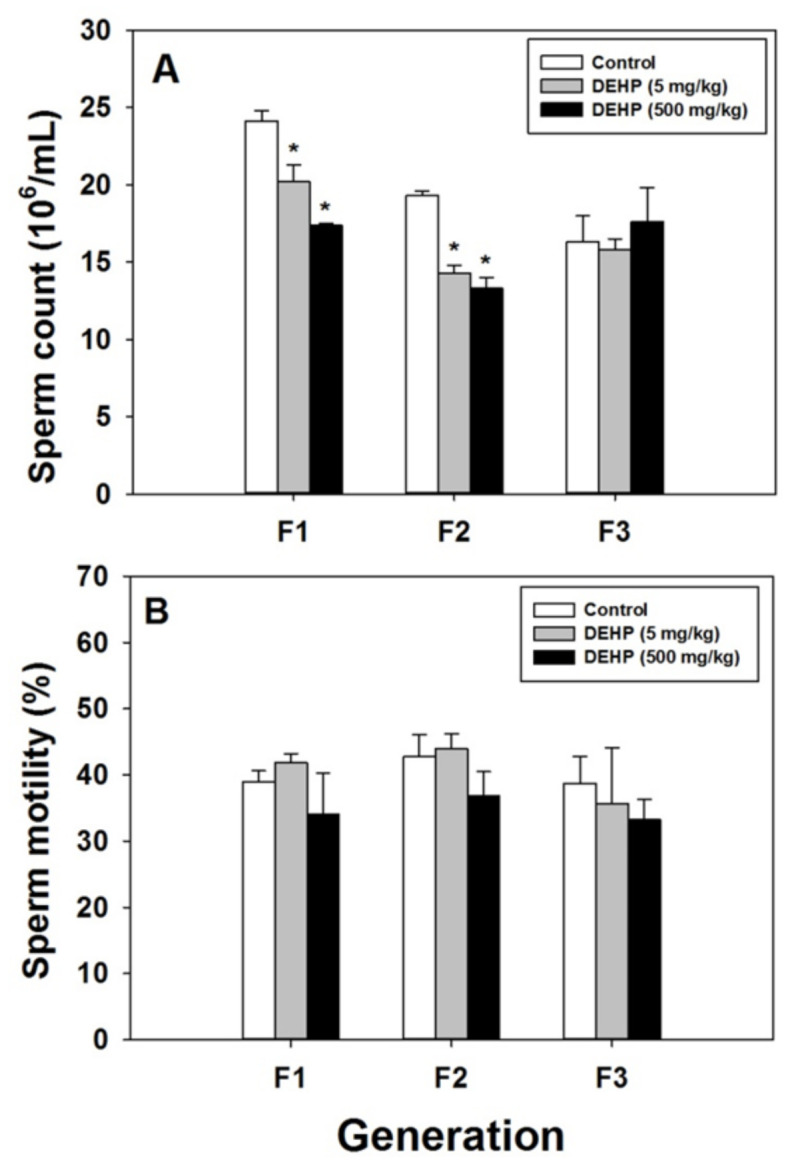
Effects of prenatal exposure to DEHP on sperm count (**A**) and sperm motility (**B**) in the F1 to F3 generations of male offspring rats. Error bars represent the standard error. * indicates a significant difference with the control group (*p* < 0.05).

**Figure 5 ijms-22-04131-f005:**
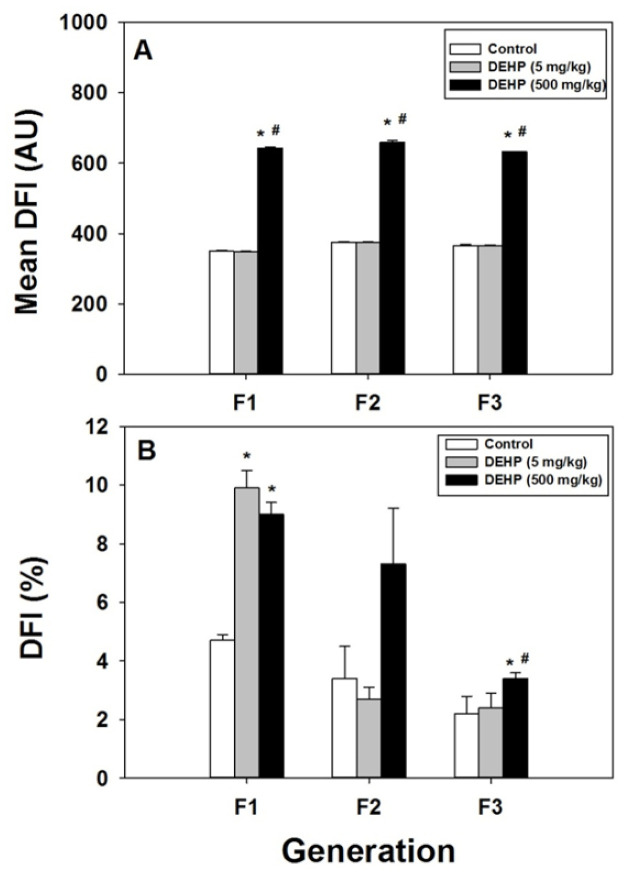
Effects of prenatal exposure to DEHP on mean DFI (**A**) and %DFI (**B**), determined using a sperm chromatin structure assay, in the F1 to F3 generations of male offspring rats. Error bars represent the standard error. * indicates a significant difference with the control group (*p* < 0.05); # indicates a significant difference with the 5 mg/kg DEHP group (*p* < 0.05).

**Figure 6 ijms-22-04131-f006:**
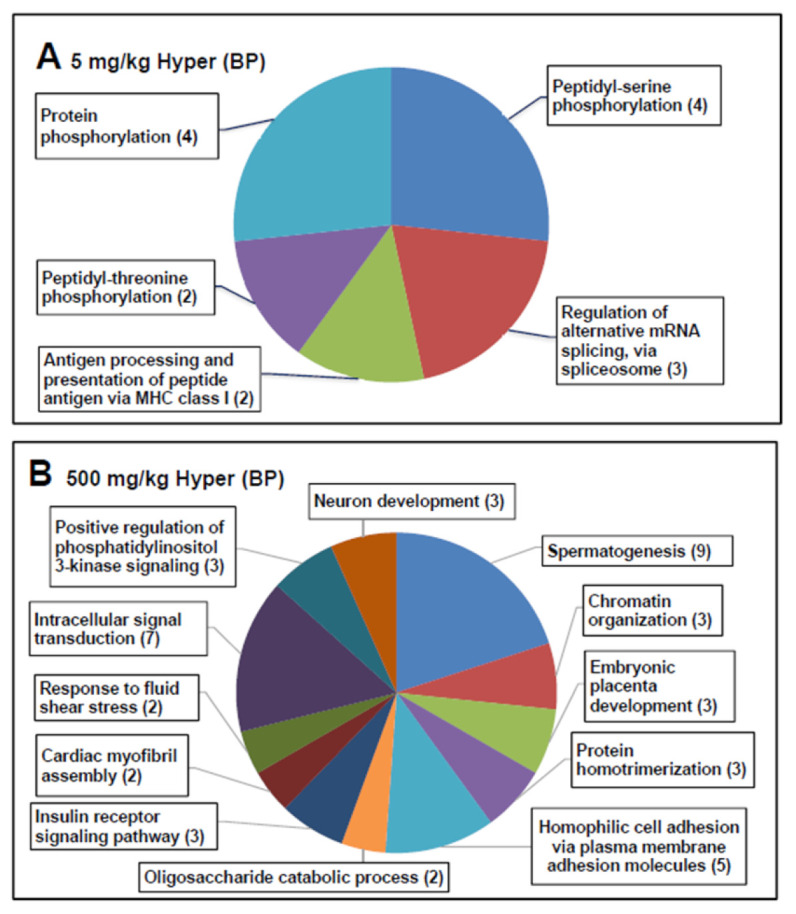
In total, 15 and 45 differentially hypermethylated genes were identified in the 5 mg/kg DEHP group (**A**) and 500 mg/kg DEHP group (**B**) relative to the control group in the F3 generation.

**Figure 7 ijms-22-04131-f007:**
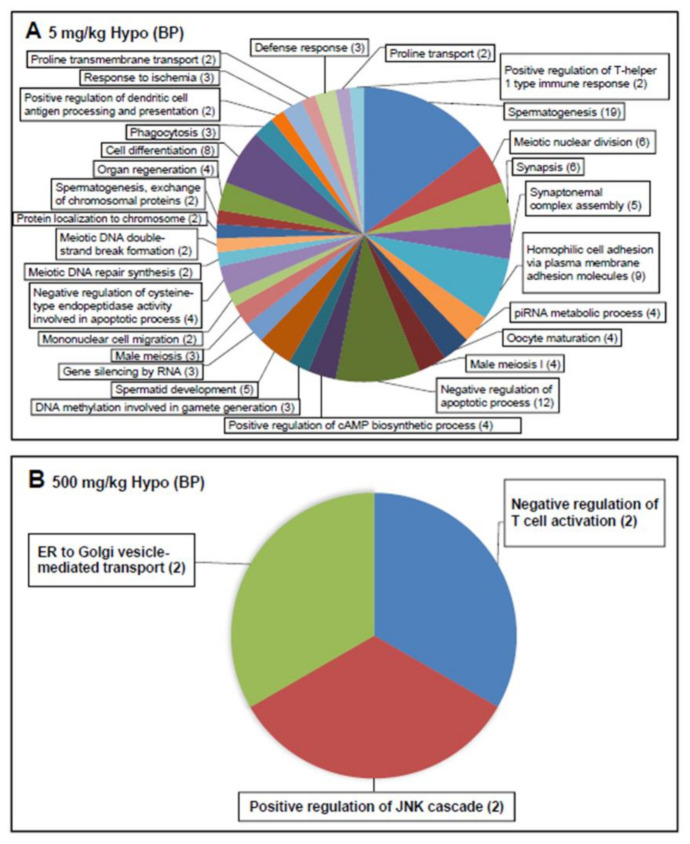
In total, 130 and 6 differentially hypomethylated genes were identified in the 5 mg/kg DEHP group (**A**) and 500 mg/kg DEHP group (**B**) relative to the control group in the F3 generation.

**Figure 8 ijms-22-04131-f008:**
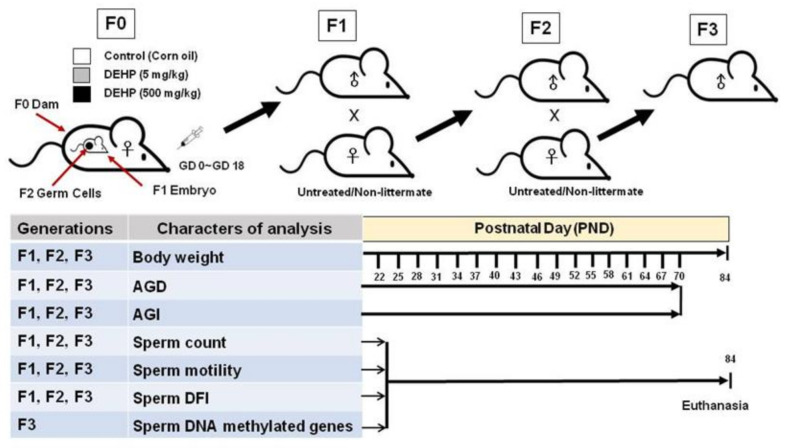
Experimental procedures in the transgenerational experimental design and the timeline of individual measurements.

**Table 1 ijms-22-04131-t001:** The 10 most hypermethylated genes in terms of relative fold change in reads per kilobase per million (RPKM) expression between the 500 mg/kg DEHP, 5 mg/kg DEHP, and control groups. After normalization (capture/input), hypermethylated genes were defined on the basis of a change of at least twofold in RPKM relative to the control group in the F3 generation.

Gene Symbol	Gene Name	Chromosome	Relative Fold of RPKM Expression between 5 mg/kg and Control Groups *	Relative Fold of RPKM Expression between 500 mg/kg and 5 mg/kg Groups ^#^
*FAM222A*	Family with sequence similarity 222, member A	Chr12	4.44	2.32 (10.30) ^a^
*GAPDH*	Glyceraldehyde-3-phosphate Dehydrogenase	Chr4	4.23	1.86 (7.87)
*CPSF2*	Cleavage and polyadenylation specific factor 2	Chr6	2.84	2.61 (7.41)
*ESRRA*	Estrogen-related receptor alpha	Chr1	3.73	1.79 (6.68)
*ANKRD13D*	Ankyrin repeat domain 13 family, member D	Chr1	4.38	1.51 (6.61)
*TTBK1*	Tau Tubulin Kinase 1	Chr9	3.79	1.72 (6.52)
*TBCCD1*	Tubulin cofactor C TBCC-domain containing 1	Chr11	4.84	1.26 (6.10)
*FAM220A*	Family with sequence similarity 220, member A	Chr12	4.55	1.33 (6.05)
*CD302*	CD302 antigen	Chr3	5.06	1.19 (6.02)
*MTMR7*	Myotubularin-related protein 7	Chr16	2.91	2.04 (5.94)

* Ratio of relative fold expression of RPKM between 5 mg/kg DEHP and control groups: (RPKM of capture/input in 5 mg/kg DEHP group)/(RPKM of capture/input in control group); ^#^ ratio of relative fold expression of RPKM between 500 mg/kg DEHP and 5 mg/kg DEHP groups: (RPKM of capture/input in DEHP 500 mg/kg DEHP group)/(RPKM of capture/input in 5 mg/kg DEHP group); ^a^ relative fold of RPKM expression between 5 mg/kg DEHP and control groups × relative fold of RPKM expression between 500 mg/kg DEHP and 5 mg/kg DEHP groups.

**Table 2 ijms-22-04131-t002:** The 10 most hypomethylated genes in terms of relative fold change in reads per kilobase per million (RPKM) expression between the 500 mg/kg DEHP, 5 mg/kg DEHP, and control groups. After normalization (capture/input), hypomethylated genes were defined on the basis of a change of less than half-fold in RPKM relative to the control group in the F3 generation.

Gene Symbol	Gene Name	Chromosome	Relative Fold of RPKM Expression between 5 mg/kg and Control Groups *	Relative Fold of RPKM Expression between 500 mg/kg and 5 mg/kg Groups ^#^
*ATG16L1*	Autophagy Atg16L1	Chr9	0.47	0.50 (0.24) ^a^
*FAM13A*	Family with sequence similarity 13, member A	Chr4	0.49	0.49 (0.24)
*ATOH7*	Atonal Homolog 7	Chr20	0.50	0.68 (0.34)
*BCAP29*	B-cell receptor-associated protein 29	Chr6	0.36	0.98 (0.35)
*MRPS18B*	Mitochondrial ribosomal protein S18B	Chr20	0.42	0.85 (0.36)
*DUSP22*	Dual-specificity phosphatases 22	Chr17	0.39	0.92 (0.36)
*AOX2*	Alcohol oxidase 2	Chr9	0.46	0.81 (0.37)
*RBM*	RNA-binding motif	Chr8	0.50	0.77 (0.39)
*MED24*	Mediator complex subunit 24	Chr10	0.42	0.92 (0.39)
*DDX19A*	DEAD box polypeptide 19A	Chr19	0.46	0.85 (0.39)

* Ratio of relative fold expression of RPKM between 5 mg/kg DEHP and control groups: (RPKM of capture/input in 5 mg/kg DEHP group)/(RPKM of capture/input in control group); ^#^ ratio of relative fold expression of RPKM between 500 mg/kg DEHP and 5 mg/kg DEHP groups: (RPKM of capture/input in DEHP 500 mg/kg DEHP group)/(RPKM of capture/input in 5 mg/kg DEHP group); ^a^ relative fold of RPKM expression between 5 mg/kg DEHP and control groups × relative fold of RPKM expression between 500 mg/kg DEHP and 5 mg/kg DEHP groups.

## Data Availability

The data presented in this study are available on request from the corresponding author.

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
