# Peer review of "Transgenerational Effects of Di(2-Ethylhexyl) Phthalate on Anogenital Distance, Sperm Functions and DNA Methylation in Rat Offspring"

_ijms, 2021, doi:10.3390/ijms22084131_

Round 1

Reviewer 1 Report

I have several major concerns, which should be considered.

  • The choice of the dose, the authors should justify the choice of those doses. How doses are close to environmental exposure?
  • SD model, is that Sprague-Dawley? Why was rat model used? Why not a mouse model? Mouse model is commonly used in toxicological studies.
  • Why in F3 control rats is less sperm than in F1? I could imagine that it should be opposite, F1 control were exposed to oil via gavage, which is stressful treatment and that could lead to spermatozoa decline even in F1 control.
  • The results of DNA fragmentation index are observed to be extremely high in F3. Normally, with each generation normally the induced effects are less obvious and diminished. Could authors comment on that. Could author show the row results of FACS print screen data?
  • The DNA methylation in F1 was not performed and it is not clear for me whether in F3 DNA methylation alterations were inherited from F1? Could authors comment on that?
  • Why the authors did not show the effects on testis and other organ weights? It could be informative for assessment of whether the dose is very toxic or not.
  • Material from ow many animals were pooled for each replicate for DNA sequencing analysis.
  • The authors should demonstrate the reproducibility between replicates using Spearman correlation coefficients analysis.
  • How was the statistical significance of sequencing data assessed?  
  • Does multiple testing errors was applied?
  • Which negative control samples were used for the quality control of Methylation analysis?
  • How was alignment performed considering that some reads are not unique?
  • How gene ontology was performed, considering small amounts of differential methylated regions, whether any GO terms were significantly enriched?

Minor points

  • How many total reads were obtained in DNA methylation analysis?
  • Number of replicates used in each assay should be specified.
  • The sequencing read mapping (bam, bedgraph) should be illustrated for at least one differentially methylated region.

Author Response

Author's Reply to the Review Report (Reviewer 1)

Comments and Suggestions for Authors

I have several major concerns, which should be considered.

ANS: Thank you for your attention to our manuscript, “Transgenerational Effects of Di(2-ethylhexyl) phthalate on Anogenital Distance, Sperm Functions and DNA Methylation in Rat Offspring”. We are able to address your concerns, and submit a revised manuscript for your consideration.

1. The choice of the dose, the authors should justify the choice of those doses. How doses are close to environmental exposure?

ANS 1: We administered the dose of 5 mg DEHP/kg/day which is approximately 167 times higher than the mean daily intake by the normal population (30 mg /kg/day) (Doull et al., 1999) and similar with the NOAEL (5 mg DEHP/kg/day) used by the European Union (EFSA, 2005).

2. SD model, is that Sprague-Dawley? Why was rat model used? Why not a mouse model? Mouse model is commonly used in toxicological studies.

ANS 2: Thank you for your suggestions. SD model is the Sprague-Dawley rat model. We found that the Sprague-Dawley rat model was widely used for evaluating the transgenerational effects on DEHP toxicity. Please see the related articles listed in the References (Manikkam et al., 2013; Chen et al., 2015; Yuan et al., 2017; Van Cauwenbergh et al., 2020).

3. Why in F3 control rats is less sperm than in F1? I could imagine that it should be opposite, F1 control were exposed to oil via gavage, which is stressful treatment and that could lead to spermatozoa decline even in F1 control.

ANS 3: In this study, the pregnant female rats treated with DEHP or corn oil were designated the F0 generation. The offspring of the F0 control generation rats were the F1 control generation. No significant differences of sperm count were observed in the control groups among F1, F2 and F3.

4. The results of DNA fragmentation index are observed to be extremely high in F3. Normally, with each generation normally the induced effects are less obvious and diminished. Could authors comment on that. Could author show the row results of FACS print screen data?

ANS 4: We have checked the row results of FACS print screen data again. In the F1 to F3 generations, the DFI was significantly higher in the 500 mg/kg DEHP group than in the 5 mg/kg DEHP and control groups. However, the results of DNA fragmentation index are not observed to be extremely high in F3.

5. The DNA methylation in F1 was not performed and it is not clear for me whether in F3 DNA methylation alterations were inherited from F1? Could authors comment on that?

ANS 5: Thank you for your concern. In this study, we hypothesized that maternal exposure to DEHP induces epigenetic transgenerational inheritance of adult-onset adverse reproductive outcomes of male offspring. If exposed in-utero, the F3 generation would be the first generation that acquires a transgenerational phenotype. Therefore, the results of sperm DNA methylation in F3 will explain the mechanism of DEHP induces epigenetic transgenerational inheritance.

6. Why the authors did not show the effects on testis and other organ weights? It could be informative for assessment of whether the dose is very toxic or not.

ANS 6: We focused on the outcomes of development, sperm functions and sperm DNA methylation. Therefore we did not show the effects on testis and other organ weights this time.

7. Material from how many animals were pooled for each replicate for DNA sequencing analysis.

ANS 7: Three experimental pools were generated for the control and DEHP-treated lineages in the F3 generation; each pool contained sperm-extracted DNA from nine animals each from three litters. Please see the Section of 4.8. Methyl-CpG binding domain sequencing.

8. The authors should demonstrate the reproducibility between replicates using Spearman correlation coefficients analysis.

ANS 8: Thanks for your suggestion. One-way analysis of variance (ANOVA) was performed for each group to identify significant differences in means between control and DEHP-exposed groups.

9. How was the statistical significance of sequencing data assessed?

ANS 9: The Trimmomatic tool was used to trim or remove reads depending on their quality score. The qualified reads were aligned with the rat reference genome sequence (RN5.0), which was retrieved from the University of California San Francisco (UCSF) database by using the Burrows–Wheeler transform. The statistical significance of sequencing data assessed as p < 0.05.

10. Does multiple testing errors was applied?

ANS 10: We did not use the method of multiple testing errors in this study.

11. Which negative control samples were used for the quality control of Methylation analysis?

ANS 11: The data of Control group were used for the quality control of Methylation analysis.

12. How was alignment performed considering that some reads are not unique?

ANS 12: The methylated DNA immunoprecipitation (MeDIP) was used to analyze the enrichment scores and determine the differences in methylation between the samples based on a window size of 500 bp and to detect alignment performed considering that some reads whether not unique. An MBD/MeDIP-seq experiment should return sequences with high CpG enrichment scores. Reads within 5000 bp upstream of genes were used to calculate the reads per kilobase per million (RPKM)

13. How gene ontology was performed, considering small amounts of differential methylated regions, whether any GO terms were significantly enriched?

ANS 13: The obtained hypermethylated and hypomethylated genes were added to the Database for Annotation, Visualization, and Integrated Discovery (DAVID) for reuse; then, the genes were grouped according to the biological processes of GO. Any GO terms were significantly enriched as compare with RPKM of capture and input.

Minor points

1. How many total reads were obtained in DNA methylation analysis?

ANS 1: Almost 20,000 total reads were obtained in DNA methylation analysis.

2. Number of replicates used in each assay should be specified.

ANS 2: In this study, number of replicates used in each assay have been specified.

3. The sequencing read mapping (bam, bedgraph) should be illustrated for at least one differentially methylated region.

ANS 3: After normalization (capture/input), hyper-/hypomethylated genes were screened in the 500 mg/kg and 5 mg/kg DEHP groups. The distributions of differentially hypermethylated genes and hypermethylated of the 500 mg/kg and 5 mg/kg DEHP groups have been illustrated in Figure 6 and Figure 7.

Reviewer 2 Report

ijms-1157823

In this study, the authors show that prenatal exposure to low doses of DEHP produces transgenic epigenetic effects in established animal models and argue that such effects may explain phenotypic changes in the male reproductive system. The in vivo exposure experimental system using rats is reasonable, and the authors' demonstration of hypomethylation of a group of genes related to testis formation in the F3 generation is a new finding.

The authors believe that the following points should be revised:

  1. In the results chapter, I could not find any description of Figure 3; please describe the evaluation of AGD in the F3 generation, as it is considered to be an important data.
  2. I couldn't understand what "Table1. generation" meant in Figure legend in Figure 1.
  3. The legends of Figures 1 through 3 should show how many offsprings were averaged for each deta.
  4. In M&M, explanations for 4.5 and 4.6 are somewhat lacking. The number of sperm and the number of individuals used for sperm count calculation and motility calculation should be clearly stated; for SCSA, the FCM model, settings, and analysis methods are unclear. If omitted, references are needed.

Author Response

Author's Reply to the Review Report (Reviewer 2)

Comments and Suggestions for Authors

ijms-1157823

In this study, the authors show that prenatal exposure to low doses of DEHP produces transgenic epigenetic effects in established animal models and argue that such effects may explain phenotypic changes in the male reproductive system. The in vivo exposure experimental system using rats is reasonable, and the authors' demonstration of hypomethylation of a group of genes related to testis formation in the F3 generation is a new finding.

ANS: Thank you very much for your encouragements. In this study, we hypothesized that maternal exposure to DEHP induces epigenetic transgenerational inheritance of adult-onset adverse reproductive outcomes of male offspring. If exposed in-utero, the F3 generation would be the first generation that acquires a transgenerational phenotype. Therefore, the results of sperm DNA methylation in F3 will explain the mechanism of DEHP induces epigenetic transgenerational inheritance. We believed that the demonstration of hypomethylation of a group of genes related to testis formation in the F3 generation would be a new finding of this study.

The authors believe that the following points should be revised:

1. In the results chapter, I could not find any description of Figure 3; please describe the evaluation of AGD in the F3 generation, as it is considered to be an important data.

ANS 1: Thank you very much for your remind. We have added the description of Figure 3 regarding the evaluation of AGD in the F3 generation. Please see the 3rd paragraph of 2.1. Effects of DEHP exposure on body weight, AGD, and AGI.

2. I couldn't understand what “Table1. Generation” meant in Figure legend in Figure 1.

ANS 2: Thanks for your suggestion. We have corrected the words from “Table1. Generation” to “the F1 generation” in legend in Figure 1.

3. The legends of Figures 1 through 3 should show how many offsprings were averaged for each deta.

ANS 3: We have added the sentence of “Data from three litters were used for every treatment group” to show how many offsprings were averaged for each deta in the legends of Figures 1 through 3.

4. In M&M, explanations for 4.5 and 4.6 are somewhat lacking. The number of sperm and the number of individuals used for sperm count calculation and motility calculation should be clearly stated; for SCSA, the FCM model, settings, and analysis methods are unclear. If omitted, references are needed.

ANS 4: We have added the explanations of “motility was expressed as the ratio between the number of motile sperm and the total number of sperm” for 4.5 and “AO is used to distinguish stained double-stranded nucleic acids from those that are single-stranded. In SCSA analysis using BD FACS-Canto II FCM (BD Biosciences, San Jose, CA, USA), each AO-stained spermatozoon DNA analyzed produced green fluorescence (detected by a 515–530 nm band-pass filter) and red fluorescence (detected by a 630–650-nm long-pass filter). Sperm DNA DFI and %DFI were analyzed according to Evenson and Wixon (2005) [42]” for 4.6 to describe the methodologies of sperm motility and SCSA in details.